# In Situ Medium Entropy Intermetallic Reinforced Composite Coating Fabricated by Additive Manufacturing

**Ahmad Ostovari Moghaddam [1]**, **Nataliya Aleksandrovna Shaburova [2]**, **Marina Nikolaevna Samodurova [3]**, **Yuliya Sergeevna Latfulina [3]**, **Dmitry Vyacheslavovich Mikhailov [2]** and **Evgeny Alekseevich Trofimov [2,\*]**

1 Advanced Materials and Nanotechnology Research Laboratory, Faculty of Materials Science and Engineering, K.N. Toosi University of Technology, Tehran 1999143344, Iran; ostovary@aut.ac.ir
2 Department of Materials Science, Physical and Chemical Properties of Materials, South Ural State University, 76 Lenin Av., 454080 Chelyabinsk, Russia; shaburovana@susu.ru (N.A.S.); a_b_c81@mail.ru (D.V.M.)
3 Department of Information and Measuring Technology, South Ural State University, 76 Lenin Av., 454080 Chelyabinsk, Russia; samodurovamn@susu.ru (M.N.S.); latfulinays@susu.ru (Y.S.L.)
\* Correspondence: trofimovea@susu.ru; Tel.: +8-(351)-272-32-10

**Abstract:** The possibility of stabilizing different amounts of medium-entropy intermetallic compounds (MEIMCs) within a multicomponent matrix using laser cladding is demonstrated. The results indicated that MEIMC with a B2 structure could be successfully formed within a multicomponent BCC matrix during laser cladding of a proper ratio of Al, Fe, Co, Cu, Mn, and Ni powders. Two coatings with different contents of MEIMC were fabricated by changing the feeding rate of the powder mixture. Based on the X-ray diffraction (XRD) and energy dispersive spectroscopy (EDS) analyses, the Al-rich intermetallic particles were qualitatively identified as $(Fe_{0.55}Co_{0.18}Cu_{0.22}Ni_{0.03}Mn_{0.02})Al$ MEIMC. It was also found that the feeding rate affects the content of MEIMC, and consequently, the grain structure and microhardness values. Finally, we propose MEIMC-reinforced alloys as a more effective alternative system to be used for fabricating high-performance coatings using laser cladding.

**Keywords:** medium-entropy intermetallic compounds; additive manufacturing; crystal structure; microhardness

## 1. Introduction

The development of high entropy alloys (HEAs), which typically consisted of five or more principal elements with (near) the equiatomic ratio, has opened the door to a vast range of feasible compositions to fabricate novel materials with superior properties [1–3]. While HEAs being solid solution alloys, recently, several high entropy intermetallic compounds (HEIMCs) have been fabricated, bridging the conventional HEAs and the emerging non-metallic high entropy materials [4–6]. The synthesis of single-phase HEIMCs was first reported by Mishra et al. [7] in 2018, where they synthesized hexanary TiZrVCrNiFex (x = 0.5, 1.0, 1.5) intermetallic compounds (IMCs) having C14 Laves phase.

All of the as-cast and rapidly quenched alloys possess the C14 Laves phase, which was stable up to 800 °C. It was found that Zr and Ti atoms occupy the A site, while the other four constituent elements (V, Cr, Ni, and Fe) fill the B site in the lattice, creating a pseudobinary AB2-type structure. They successfully demonstrated that the stability of the Laves phase HEIMC well correlated with the thermodynamic calculations following Miedema's approach and the structural parameter descriptors such as mixing enthalpy, configurational entropy, electronegativity, atomic size mismatch, and valence electron concentration. Unfortunately, the authors just investigated the structural properties of these HEIMCs and did not report any data about their mechanical or functional properties. The second work on HEIMCs was reported in 2019, in which Zhou et al. [6] proposed a new criterion for forming B2 single-phase HEIMCs and developed several multicomponent B2 single-phase aluminides such as $(Co_{1/4}Fe_{1/4}Ni_{1/4}Mn_{1/4})Al$, $(Co_{1/4}Fe_{1/4}Ni_{1/4}Cu_{1/4})Al$,

and $(Fe_{1/5}Co_{1/5}Ni_{1/5}Mn_{1/5}Cu_{1/5})Al$. Typically, IMCs exhibit superior thermal stability and high-temperature strength compared to the solid solution alloys [8]. These criteria may likely be synergized in HEIMCs owing to the intrinsic features of the entropy stabilized phases, as recently realized by Yao et al. [5] in $(Fe_{1/4}Co_{1/4}Ni_{1/4}Cu_{1/4})$ $(Ti_{1/3}Zr_{1/3}Hf_{1/3})$ HEIMC with a single B2 structure. This HEIMC demonstrated exceptional high-temperature strength exhibiting yield strengths of 905 MPa at 800 °C and 705 MPa at 900 °C.

In addition to the single-phase HEIMC, it is also demonstrated that HEIMC could be utilized as nanoparticles for strengthening in metallic systems. For example, Yang et al. [9] developed a route to break the strength–ductility trade-off by the in situ creation L12-HEIMC nanoparticles for coherent strengthening in the FeCoNi-base fcc HEA systems. The L12-type HEIMC with a composition $(Ni_{43.3}Co_{23.7}Fe_8)_3$ $(Ti_{14.4}Al_{8.6}Fe_2)$ was successfully developed, which had a significant effect on the mechanical properties of the alloy. They developed an alloy with superior strengths of 1.5 GPa and ductility as high as 50% in tension at room temperature. This superior behavior was attributed to the noticeable dislocation activities and deformation-induced microbands and the ensuing multistage work-hardening characteristic of the material. It should also be noted that several works have been published on the reinforcing of HEAs with conventional intermetallic compounds (not HEIMCs) [10,11].

Tsai et al. [12] noted the complexity of the creation of intermetallic (IM) phases in HEAs and used a statistical approach to provide a fundamental understanding of the IM phases formed in HEAs. Their results showed that the five most common IM structures in the 142 HEAs examined are the Laves, σ, B2, L1$_2$, and L2$_1$ phases. With regard to structural inheritance, all IM phases contained in the alloys represent existing structures in the binary/ternary subsystems of the respective alloys. They concluded that the complexity of composition in HEAs induces additional complexity in the formation of the IM structure.

While still being in their infancy, HEIMCs have been synthesized by conventional arc melting or mechanical alloying. However, additive manufacturing (AM) presents a unique opportunity to fabricate complicated parts and coatings [13,14]. Especially, laser cladding is a well-developed AM technique to deposit coatings from a wide range of materials. The manufacturing of HEAs using AM was previously demonstrated [13]. However, to the best of our knowledge, the possibility of AM of entropy stabilized IMCs has not been demonstrated. In the absence of any report on the AM of entropy stabilized IMCs, this work demonstrates the possibility of in-situ obtaining medium-entropy IMCs (MEIMCs) as a strengthening phase in a multicomponent alloy coating using direct laser cladding.

## 2. Materials and Methods

To create the coating, a mechanical mixture of pure metal powders (GTV GmbH, Luckenbach, Germany) (spherical particles with sizes of about 50 micrometers) was used. After several tests, and considering the Fe-induced dilution effect from the substrate, we adjusted the chemical composition of the starting powders to obtain a desirable amount of MEIMC in the Fe-based matrix. The following composition was used to fabricate the coatings (at.%): 50 Al, 10 Cu, 10 Co, 2.5 Ni, 25 Fe, and 2.5 Mn. Low-alloyed steel used as a substrate had the composition (wt.%): 0.34% C, 0.38% Si, 0.68% Mn, 1.47% Cr, 1.53% Ni, 0.25% Mo, 0.017% S, <0.035% P, and bal. Fe. The substrate material was used in the annealed state.

To form the coating, a laser metal-cladding unit FL-Clad-R-4 was used (IPG IRE-Polus, IPG Photonics Corporation, Fryazino, Russia). The main parts of the unit can be represented as follows (Figure 1): (1) 4 kW laser head with an ytterbium fiber-optics laser with wavelength 1065–1075 nm (LS-4), the heating mode is continuous; (2) KUKA R-120 six-axis robot-equipped manipulator combined with KUKA DKP-400 double-axis positioning element; (3) TWIN-10-CR-2 powder feeder with a four-axis powder feed module; (4) process chamber–a metal cylinder with the diameter of 600 mm and the length of 1100 mm. The trajectory of the laser path was linear. Surfaced strips of coating partially

overlapped each other. The overlap width was about 1.2 mm. The coating was carried out under shielding gas (argon). Two experiments were carried out, fundamentally differing in the powder feed rate of 6 g/min and 9 g/min. The technological parameters of the processes are given in Table 1.

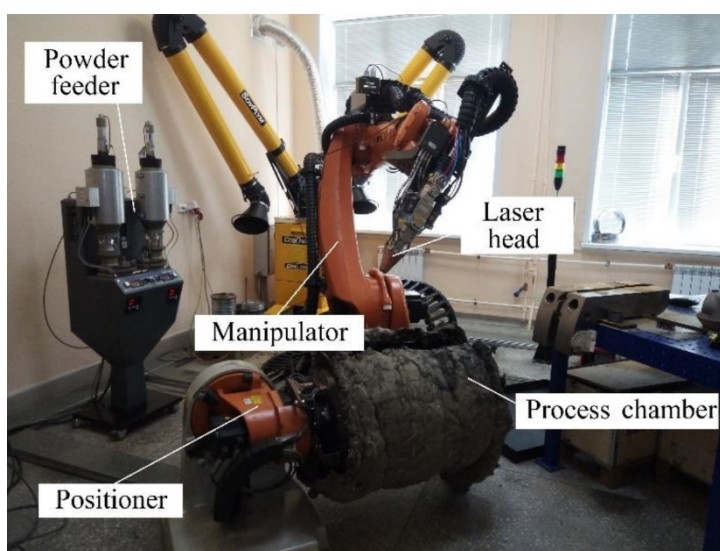

**Figure 1.** The laser metal-cladding unit FL-Clad-R-4.

**Table 1.** Technological parameters of the experiments.

| Samples | Laser Power, W | Rate of Deposition, mm/sec | Gas Flow Rate, L/min | Powder Mass Flow, g/min | Spot Diameter, mm | Track Offset, mm |
|---|---|---|---|---|---|---|
| Sample-6 | 1200 | 12 | 7 | 6 | 3 | 1.4 |
| Sample-9 | 1200 | 12 | 7 | 9 | 3 | 1.4 |

X-ray diffraction (XRD) was performed on a Rigaku Ultima IV X-ray diffractometer (Rigaku, Tokyo, Japan) using Cu–Kα radiation (λ = 0.154 06 Å) to reveal crystal structures of the coating. The microstructure was studied on cross-sections of the samples using an Axio Observer D1.m inverted optical metallographic microscope (Carl Zeiss Microscopy GmbH, Jena, Germany) equipped with Thixomet Pro software and hardware complex (Thixomet Pro, Thixomet Company, St. Petersburg, Russia) for image analysis and grain size calculation. The microstructure of the samples was also studied using a JEOL JSM-7001F scanning electron microscope (SEM) (JEOL, Tokyo, Japan) with Oxford Instruments energy dispersive x-ray spectrometer (EDS) (Oxford Instruments, Abingdon, UK) for quantitative and qualitative X-ray microanalysis (XRMA). For microstructural studies, the samples were etched in Kalling's solution (5 g of copper chloride, 100 mL of hydrochloric acid (density 1.19), 100 mL of ethyl alcohol, 100 mL of distilled water). The microhardness was measured on cross-sections in the direction from the surface (coating) to the substrate using an FM-800 microhardness tester (Future-Tech Corp, Kawasaki, Japan). A load of 300 g for 10 s dwell time was employed. The distance of indenter from the edge of the section was not less than 1.5 times of the printed diagonals' diameter, and the distance between the adjacent measurement points was not less than two diagonals of the indenter prints.

## 3. Results

### 3.1. Microstructure and Phase Analyses

Figure 2 shows the XRD spectra of the cladded coatings. Le Bail refinement was performed to calculate the lattice parameters of the samples. The XRD spectra of both coatings indicated the presence of the BCC and B2 (a BCC structure with CsCl ordering) phases. It should be noted that the B2 phase has the same diffraction peaks of (110), (200)

and (211) planes of BCC. However, the B2 phase also exhibits the diffraction peaks of (110, (111), and (210) planes which are the characteristic peaks of the superlattice structure, and confirms the formation of B2 phase. These are forbidden peaks for disordered BCC structures due to the symmetry. The XRD spectra matched well with a mixture of only disordered BCC solid solution and ordered B2 IMC for both coatings. No other phases could be detected in the XRD spectra of the samples. The calculated phase compositions and lattice parameters of these phases for both samples are shown in Table 2.

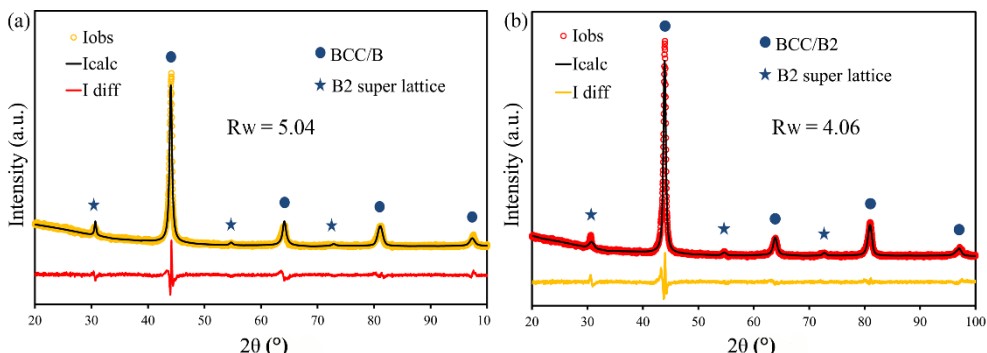

**Figure 2.** XRD spectra of the coatings fabricated using a feeding rate of (**a**) 6 g/min and (**b**) 9 g/min. The weighted-profile R-factor (Rw) is also indicated.

**Table 2.** EDS chemical compositions measured from the cross-section of the fabricated coating. The average values of five measurements are reported.

| Samples | Feeding Rate (g/min) | EDS Chemical Composition (at. %) | | | | | | Phase Fractions (%) | Lattice Parameter (Å) | |
|---|---|---|---|---|---|---|---|---|---|---|
| | | Al | Co | Cu | Fe | Ni | Mn | | BCC | B2 |
| Sample-6 | 6 | 20.42 | 5.52 | 6.73 | 66.03 | 0.52 | 0.78 | 66.2 BCC + 33.8 B2 | 2.911 | 2.902 |
| Sample-9 | 9 | 35.66 | 7.61 | 10.26 | 44.25 | 1.25 | 0.97 | 39.9 BCC + 60.1 B2 | 2.915 | 2.907 |

Figures 3 and 4 show the microstructure of the coatings fabricated by using the feed rates of 6 g/min (sample-6) and 9 g/min (sample-9), respectively. The contact zone of the substrate with the coating for both samples has no visible cracks, pores and other discontinuities. Figure 3 shows the microstructure of the coating obtained at a low powder feed rate of 6 g/min. It can be seen that the coating has dimensions of 1.2 and 0.8 mm along the axis of the deposited beads and in the overlapped zones, respectively. The structure of the coating metal is uniform over the thickness, which is characterized by large columnar grains with sizes ranging from 100 to 800 μm. The high magnification SEM image in Figure 3b clearly indicates the presence of intermetallic precipitates inside the grains and along the grain boundaries. EDS elemental maps indicated that these precipitates are Al-rich and Fe-poor, while the other elements have a homogenous distribution.

The coating fabricated at a higher feeding rate (9 g/min) exhibited an entirely different microstructure, as can be seen in Figure 4a. Similar to sample-6, the melting baths formed during the laser cladding are clearly visible on the cross-section after etching. The thickness of the coating along the deposited beads is about 800 μm, while in the overlapped zones is below 600 μm. The structure of the coating material is heterogeneous in thickness, consisting of a 100 μm area adjacent to the substrate having equiaxed grains with a size of 50 μm, followed by a region with a thickness of 350–400 μm and fine equiaxed grains of about 25 μm. The main part of the coating has an equiaxed fine-grained structure with a grain size of no more than 25 μm (see Figure 4a,b). Similar to sample-6, a high density of intermetallic precipitates are observed along the boundaries and inside the grains (Figure 4b). The elemental maps indicated that Ni, Mn, Cu, and Co have a uniform distribution inside the grains, while segregations are observed for Al and to some extent for Fe, indicating the formation of Al-rich intermetallic precipitates.

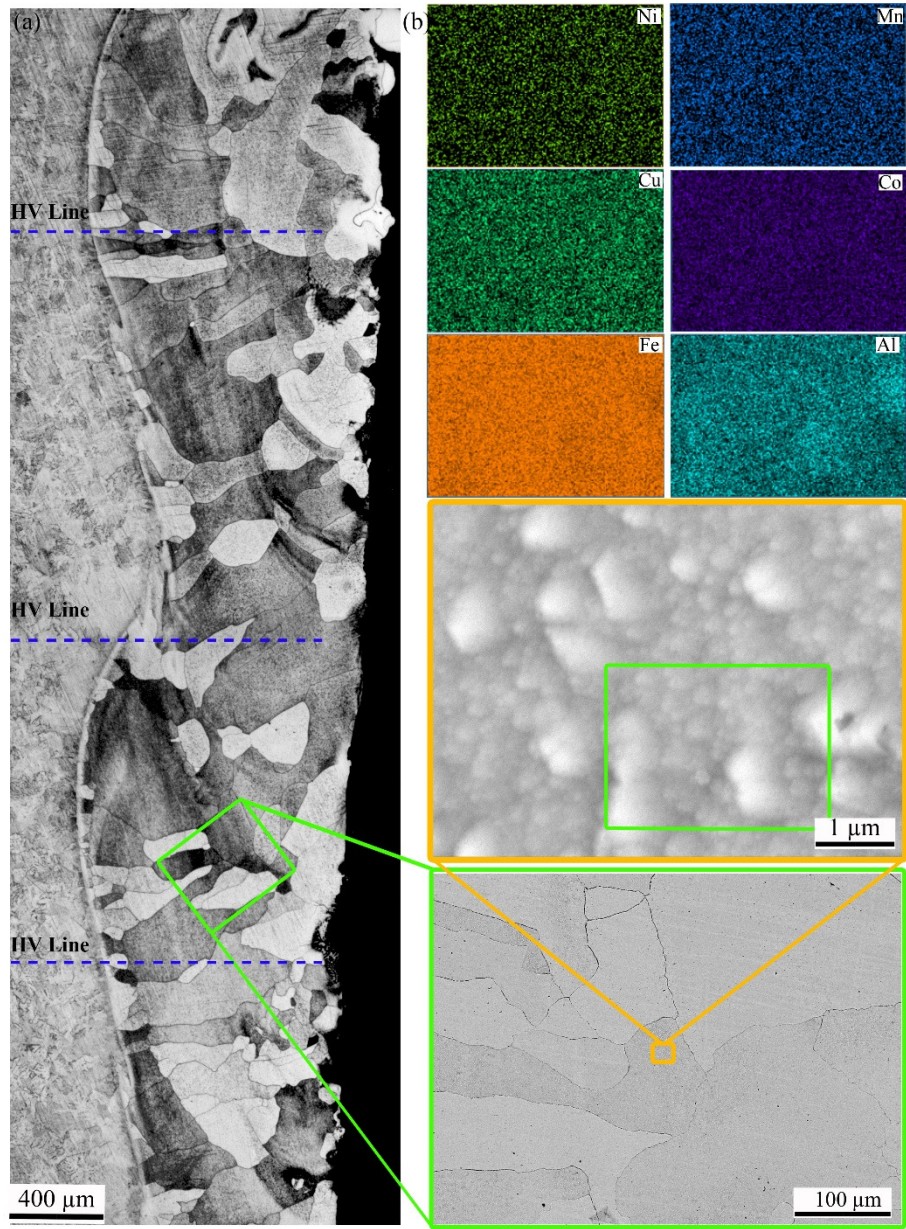

**Figure 3.** Microstructure and maps of elements distribution in the coating (sample-6): (**a**) optical image, (**b**) SEM images and the corresponding elemental maps. The dashed lines in Figure 3a show the paths of microhardness measurement.

The average chemical composition of the coatings is presented in Table 2. A clear dilution of the initial chemical composition can be detected for both coatings. This was especially the case when the cladding process was carried out at a lower feeding rate (6 g/min), a relatively high amount of Fe was incorporated into the coating from the substrate. Considering the interaction volume of electrons with the sample in EDS analysis (which is about 1 μm [15]), the precise chemical composition of the precipitates could not be determined in both samples. However, based on the XRD results, it can be concluded that the microstructure of both samples consisted of BCC micro-grains and B2 precipitates inside the BCC grains and along the grain boundaries. Moreover, it can also be concluded that both phases are multicomponent or medium-entropy systems.

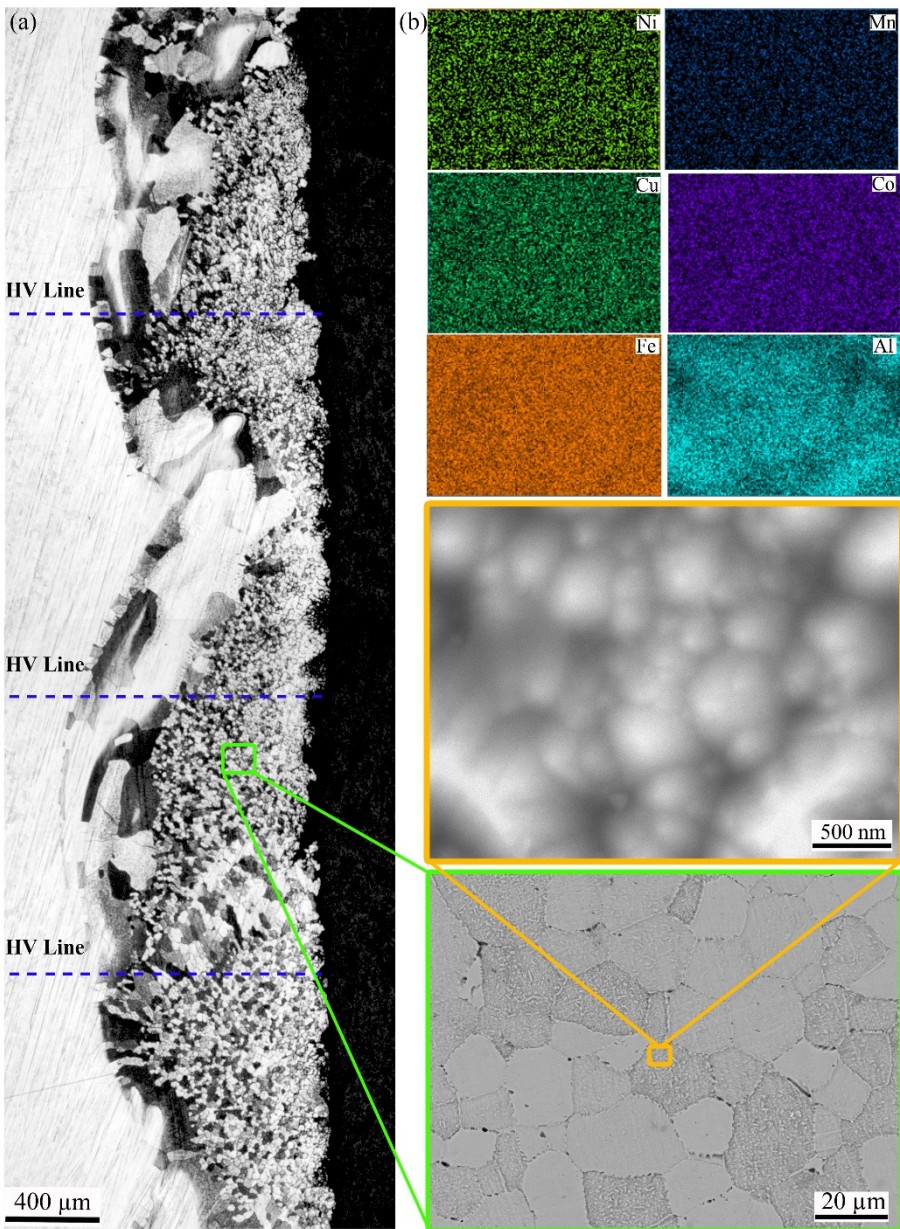

**Figure 4.** Microstructure and maps of elements distribution in the coating (sample-9): (**a**) optical image, (**b**) SEM images and the corresponding elemental maps. The dashed lines in Figure 4a show the paths of microhardness measurement.

### 3.2. Results of Microhardness Test

The microhardness of both samples measured in the cross-section along the thickness of the coating is shown in Figure 5. It can be noted that the steel substrate has a low hardness in the order of 130–150 HV. Significantly higher microhardness values were obtained for both coatings even near the substrate-coating interface. For sample-6, the structure was coarse but rather uniform. As a result, the coating exhibited close microhardness values along the entire length of its thickness, increasing from 370 HV near the substrate to about 430 HV near the surface. On the other hand, structural heterogeneity of the sample-9 coating affects its mechanical properties. The microhardness of sample-9 near the substrate (in the coarse-grained region) is about 100 HV lower than that near the surface of the coating (in the fine equiaxed grain region). The hardness of the coating metal is 3–3.5 times higher than the hardness of the substrate.

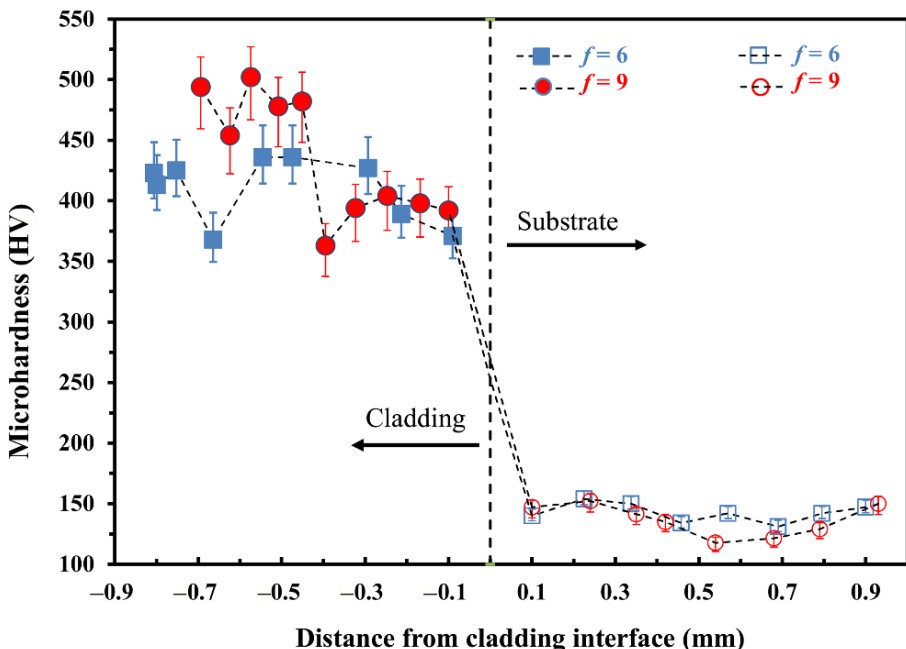

**Figure 5.** The results of microhardness measurement for the coatings.

## 4. Discussion

A relatively large number of works have been devoted to the use of laser cladding for fabricating HEA coatings [16–19]. These works demonstrated the possibility of obtaining HEA coatings of various compositions by additive manufacturing. However, the in situ formation of entropy stabilized IMCs by additive manufacturing has not been demonstrated before this work. In this work, we demonstrated for the first time that entropy-stabilized IMCs can be formed in situ during laser cladding with proper adjustment of the chemical composition of the precursor powders. By highly alloying with the Al addition and proper feeding rate, we successfully induced high-density B2 intermetallic particles in a Fe-base multicomponent alloy coating using laser cladding.

A mixture of Al-Co-Cu-Fe-Mn-Ni powders typically crystallizes in a single FCC structure at low Al contents, a duplex FCC + BCC structure at intermediate Al contents, and finally a single BCC structure at high Al concentrations [20]. The formation of a B2 intermetallic phase could only be triggered at the sufficiently high Al contents, which satisfy the criteria of AB composition. In this study, the coatings fabricated by direct laser cladding with different feeding rates exhibited different amounts of evolved IMCs and, consequently, different microstructural features. When the feeding rate was low (6 g/min), a small amount of B2-IMCs was evolved, and the microstructure consisted of elongated BCC columnar grains. The content of IMCs significantly increased when the feeding rate was increased to 9 g/min. This high content of IMCs hindered grain growth and a fine, equiaxed grain structure prevailed. For both samples, the IMCs formed inside and along the grain boundaries of the BCC phase. The difference is observed in the quantitative ratio of these phases and the final chemical composition of the resulting coating. At a low powder feed rate (6 g/min), the bcc phase predominates in the structure (its content was about 66.2%,), while by increasing the feeding rate to 9 g/min, the bcc phase decreases to about 40%. Therefore, there is a clear correlation between the grain size in the coating and the amount of the intermetallic phase. The amount of B2 phase in the sample is directly correlated with the amount of the IMC-forming elements (mainly Al, Cu, Co, and Fe) in the molten bath. More IMC-forming elements are incorporated into the molten bath at higher feeding rates, which evolves higher amounts of IMCs. It is obvious that the formation of the intermetallic and BCC phases occurs almost simultaneously.

The lattice parameters of both BCC and B2 phases in sample-9 are almost identical to that of sample-6 (Table 2). The lattice parameter of BCC phases are higher than the lattice parameter of Fe (2.87 Å [21]), which may be attributed to the substitution of Al, Cu, and Co elements with higher atomic radius than Fe in the BCC structure. This coupled with the EDS analysis suggesting that the BCC phase is a Fe-rich multicomponent solid solution. Similarly, the lattice parameter of B2 phase in both alloys is higher than the lattice parameter of binary aluminides [4] and close to those reported for multicomponent aluminides [4,6].

Here, the chemical composition of BCC and B2 phases could not be precisely determined considering the size of IMCs and the interaction volume of electrons with the sample in EDS. We could detect clear elemental segregation only for Al, while Fe exhibited a slight enrichment in the matrix, and other elements were uniformly distributed in the alloy. These Al-rich precipitates were clearly an entropy-stabilized B2 intermetallic phase. Based on the XRD and EDS analysis, the resulting intermetallic phase could be qualitatively designated based on the binary FeAl as $(Fe_{0.55}Co_{0.18}Cu_{0.22}Ni_{0.03}Mn_{0.02})Al$ MEIMC. The higher amount of Fe compared to the initial composition could be attributed to the dilution effect of the steel substrate. It is known that dilution has a detrimental effect on the chemical and mechanical properties of the materials, while a minimum dilution is required for adequate metallurgical bonding [22,23]. However, our results indicate that the dilution effect (Fe-rich) may also be used for in situ fabrication of MEIMC-reinforced composites. In this MEIMC, the Fe sublattice is substituted with several transition elements that randomly distribute over the Fe sublattice. Therefore, it can be assumed that Al atoms occupy the body-centered positions, and Fe, Co, Cu, Ni, and Mn randomly fill the Fe sites of the B2-FeAl prototype structure based on their available ratio.

The comparison of our results with the literature confirms that the chemical composition and the crystallization conditions of a high-entropy system play a decisive role in the formation of the final structures. A most prominent route may be to obtain not a single-phase HEIMC, but a composite material reinforced with HEIMC/MEIMC.

With regard to the microhardness results, the MEIMC-reinforced coatings exhibited comparable values to those of BCC HEAs fabricated by laser cladding [24,25]. In our samples, the hardness of the deposited coatings varies from 400 to 500 HV and correlates with the grain size of the coating. Considering the susceptibility of HEAs for cracking during additive manufacturing, we propose HEIMC/MEIMC-reinforced alloys as a more practical alternative system to fabricate high-performance coatings for different applications.

## 5. Conclusions

In conclusion, the research results showed that additive manufacturing can be successfully employed to fabricate HEIMC/MEIMC-reinforced coatings. However, to stabilize the HEIMC/MEIMC particles, it is necessary to strictly select the chemical composition of the precursor powders and the processing conditions. In this study, the laser cladding process was carried out at two different feeding rates of 6 g/min and 9 g/min. In both cases, the MEIMC particles with a B2 structure were successfully stabilized within a multicomponent BCC matrix. The lattice parameters of both BCC and B2 phases in the samples are almost identical. The ratio of B2-MEIMC particles could be controlled by using the powder feeding rate, and it increased from about 40 wt.% to 60 wt.% when the feeding rate increased from 6 g/min to 9 g/min. Based on the XRD and EDS analyses, the intermetallic phase could be formulated as $(Fe_{0.55}Co_{0.18}Cu_{0.22}Ni_{0.03}Mn_{0.02})Al$ MEIMC. By increasing the content of the intermetallic particles, the grain size of the BCC matrix was significantly refined (from 100–800 μm to 25 μm), and the microhardness values of the coating were increased (on average from 425 HV to 475 HV). Finally, stabilizing different HEIMC/MEIMC particles in multicomponent alloys is proposed as a novel direction for additive manufacturing of HEAs.

**Author Contributions:** Conceptualization, A.O.M., N.A.S. and M.N.S.; methodology, E.A.T. and D.V.M.; validation, A.O.M., N.A.S. and E.A.T.; formal analysis, A.O.M.; investigation, A.O.M., Y.S.L. and D.V.M.; writing—original draft preparation, A.O.M.; writing—review and editing, A.O.M., N.A.S.; project administration, E.A.T. and M.N.S. All authors have read and agreed to the published version of the manuscript.

**Funding:** The research was funded by RFBR and Chelyabinsk Region, project number 20-43-740020.

**Institutional Review Board Statement:** Not applicable.

**Informed Consent Statement:** Not applicable.

**Data Availability Statement:** The raw/processed data required to reproduce these findings are available from the corresponding author upon reasonable request.

**Conflicts of Interest:** The authors declare no conflict of interest.

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
