# Peer review of "In Situ Medium Entropy Intermetallic Reinforced Composite Coating Fabricated by Additive Manufacturing"

_metals, doi:10.3390/met11071069_

Round 1
Reviewer 1 Report
Authors demonstrate the possibility of in-situ obtaining medium entropy intermetallic compounds as strengthening phase in a multicomponent alloy coating using direct laser cladding. Paper shows interesting and yet novel results that are of interest to surface engineering people. Authors provided solid scientific support to obtained results and presented them in a clear manner.
In my opinion paper has high scientific value and should be of interest to material scientists.
It can be published after small following issues have been corrected:
1. Page 1 Line 20 - Please change "was" into " were"
2. Page 2 Line 51 - Please change "This" into "These"
3. Please change "Hv" into "HV" throughout the manuscript - this is standarized use of the Vickers Hardness unit.
4. Can Authors please indicate the position of hardness measurements? Due to the "wavy" shape of the substrate/coating interface it is important to know the location of hardness lines with respect to Figures 3a and 4a.
Author Response
Dear Reviewer, we thank you for the interest and attention shown to our work and for the comments you made. They will certainly improve our work.
We also want to respond to your comments. All changes in the text are highlighted in color.
- Page 1 Line 20 - Please change "was" into " were"
Answer: Thank you, we have made changes.
- Page 2 Line 51 - Please change "This" into "These"
Answer: Thank you, we have made changes.
- Please change "Hv" into "HV" throughout the manuscript - this is standarized use of the Vickers Hardness unit.
Answer: Thank you, we have made changes.
- Can Authors please indicate the position of hardness measurements? Due to the "wavy" shape of the substrate/coating interface it is important to know the location of hardness lines with respect to Figures 3a and 4a.
Answer: The microhardness was measured on the cross-section in the transverse direction from the coating surface to the substrate in the bottom, middle and top of the cross section. We indicated the positions of microhardness measurements in Figs. 3a and 4a.

Reviewer 2 Report
The submitted manuscript entitled ‘In Situ Medium Entropy Intermetallic Reinforced Composite Coating Fabricated by Additive Manufacturing’ deals with the additive manufacturing (laser cladding) of a special coating layer. The layers themselves were investigated microstructurally and mechanically (microhardness). The manuscript is interesting and worth publishing, during its review only a few minor issues arose.
- Please remove the commercial e-mail addresses.
- In the opinion of this Rewiever, fig 3 and 4 should be evaluated for the whole depicted area. The EDS maps evaluated too small portions.
- In the hardness values, please indicate the load (NNN HV300).
Author Response
Dear Reviewer, we thank you for the interest and attention shown to our work and for the comments you made. They will certainly improve our work.
We also want to respond to your comments. All changes in the text are highlighted in color.
- Please remove the commercial e-mail addresses.
Answer: Contributors' email addresses have been changed to institutional email addresses. With the exception of D. Mikhailov, who is a master of the last year of study and does not have such an address.
- In the opinion of this Rewiever, fig 3 and 4 should be evaluated for the whole depicted area. The EDS maps evaluated too small portions.
Answer: It should be understood that you are proposing to carry out mapping from the entire area of Figures 3a and 4a. But these panoramic images were taken at 50x magnification. Mapping at such a low magnification does not allow identifying the distribution of the alloying elements at all. Besides, we want to show the segregation of Al at nanometer scale as a proof of formation of MEIMCs. Therefore, we believe that the maps of the distribution of elements should be carried out at high magnifications, as we already provided in the manuscript.
- In the hardness values, please indicate the load (NNN HV300).
Answer: In the section material and research methods, we indicated that the microhardness was measured at a load of 300 grams, since in all experiments the load was unchanged, we do not considered it necessary to introduce an additional designation to the number of hardness.

Reviewer 3 Report
The paper deals with an interesting topic related to the coating reinforcement of alloy by Additive Manufacturing. The paper is well written and presented. However, some modifications and clarifications must be done before publication. I recommend minor revision according to the following comments:
1) Page 3 line 121: Have you followed the relevant standards for the hardness measurements? If so, please specify. Specify also all information about the measurements procedure such as the application time of the load, the distance between measurements, and distance from the edge of the section.
2) Page 4 line 147. How do you evaluate the grain size? Please, if you follow the relevant standard it is recommended to report it in the text.
3) Page 4-5 line 159 "The structure of the coating material.... grain size of no more than 25 μm.". The cited paragraph seems not clear. Looking at fig. 3, it is difficult to recognize such an average value. Please explain by using an additional figure that shows the grains of the coating material and substrate.
4) Fig. 3): the green box at the bottom-right of Fig. 3 seems not to correspond to the small green box of the left picture. Please modify.
5) Fig. 5: I see from Fig. 5 that the hardness measurements were carried out with a distance between each other about 0.1mm. Does this distance match the suggestion of the standard ASTM E384 − 17?
6) Tables: please, reduce the font size in all tables if it is possible.
Looking forward to seeing the revised manuscript.
Best regards
Author Response
Dear Reviewer, we thank you for the interest and attention shown to our work and for the comments you made. They will certainly improve our work.
We also want to respond to your comments. All changes in the text are highlighted in color.
- Page 3 line 121: Have you followed the relevant standards for the hardness measurements? If so, please specify. Specify also all information about the measurements procedure such as the application time of the load, the distance between measurements, and distance from the edge of the section.
Answer: Thanks for the comment. The necessary additions were made to the text of the manuscript:
The microhardness was measured on cross sections in the direction from the surface (coating) to the substrate using an FM-800 microhardness tester (Future-Tech Corp, Kawasaki, Japan). A load of 300 g for 10 seconds dwell time was employed. The distance of indenter from the edge of the section was not less than 1.5 times of the printed diagonals diameter, and the distance between adjacent measurement points was not less than two diagonals of the indenter prints.
- Page 4 line 147. How do you evaluate the grain size? Please, if you follow the relevant standard it is recommended to report it in the text.
Answer: Thixomet Pro image analysis software was used to determine the grain size. Corresponding clarifications have been made to the text of the manuscript.
The microstructure was studied on cross sections of the samples using an Axio Ob-server D1.m inverted optical metallographic microscope (Carl Zeiss Microscopy GmbH, Jena, Germany) equipped with a Thixomet Pro software and hardware complex (Thixomet Pro, Thixomet Company., St. Petersburg , Russia) for image analysis and grain size calculation.
- Page 4-5 line 159 "The structure of the coating material.... grain size of no more than 25 μ". The cited paragraph seems not clear. Looking at fig. 3, it is difficult to recognize such an average value. Please explain by using an additional figure that shows the grains of the coating material and substrate.
Answer: Indeed, you are right, this description refers to Figure 4, not 3. It is to Figure 4 that we refer in the paragraph you indicated. These two regions can be detected in Fig. 4a and b. To avoid confusion, we have added another link to Figure 4 in the text. Adding a new figure to the manuscript just to show this feature which can be observed in Fig.4, affect the integrity of the manuscript.
- Fig. 3): the green box at the bottom-right of Fig. 3 seems not to correspond to the small green box of the left picture. Please modify.
We apologize for this mistake. The SEM image was tilted and made it difficult to find the corresponding area in optical micrograph. However, after careful observation, we could find the position of the green box in Fig. 3a.
- Fig. 5: I see from Fig. 5 that the hardness measurements were carried out with a distance between each other about 0.1mm. Does this distance match the suggestion of the standard ASTM E384 − 17?
Answer: When measuring microhardness, we were guided by ISO 6507-1: 2005 "Metallic materials - Vickers hardness test - Part 1: Test method". According to which the distance between adjacent prints should be at least two diagonals of the imprint in order to avoid the influence of metal work hardening.
- Tables: please, reduce the font size in all tables if it is possible.
Answer: The font size in tables has been reduced.

Reviewer 4 Report
In this paper “In Situ Medium Entropy Intermetallic Reinforced Composite Coating Fabricated by Additive Manufacturing” Authors presents possibilities of stabilizing different of medium entropy intermetallic compounds within a multicomponent matrix by laser cladding. Measurements indicated that this structures could be successfully formed within a multicomponent BCC matrix during laser cladding of a proper ratio of Al, Fe, Co, Cu, Mn, and Ni powders. Using X-ray diffraction (XRD) and energy dispersive spectroscopy (EDS) analyses the qualitative composition of the produced multicomposites was determined. Reinforced alloys produced by Authors were proposed as a more efficient and alternative system for the fabrication of high-performance coatings using laser plating.
In my opinion this paper is very good. Paper can be interesting for readers of Metals journal. The paper contains 5 figures and 2 tables – figures are legible and good quality.
English of the paper is rather good – in my opinion the language of the paper should be a little improved. I am asking for corrections by a native speaker.
I did not find major factual deficiencies in the reviewed article.
I find some mistakes for example:
- Introduction chapter – in my opinion should be correct. Authors should include new information about topic of a paper. More information based on worldwide (global) study.
- In my opinion Conclusions chapter should be a little changed. In this chapter there are no summary of all significant information obtained by the Authors and written in the chapters 2 and 3.
- In the list of references I found 5 papers of the Author of reviewed paper. Please indicate the differences in the studies presented in the cited article and their relation to the presented topic.
- Please prepare a literature review according to the guidelines of the Metals
The results obtained are interesting and promising. The manuscript can be accepted for publication in Metals journal in this form.
Author Response
Dear Reviewer, we thank you for the interest and attention shown to our work and for the comments you made. They will certainly improve our work.
We also want to respond to your comments. All changes in the text are highlighted in color.
English of the paper is rather good – in my opinion the language of the paper should be a little improved. I am asking for corrections by a native speaker.
Answer: Thank you for your comment. We have worked on the bugs.
- Introduction chapter – in my opinion should be correct. Authors should include new information about topic of a paper. More information based on worldwide (global) study.
Answer: While high entropy alloys (HEAs) have been extensively studied during the last two decades, entropy stabilized intermetallic compounds are a novel subgroup of HEAs which has been recently introduced. To the best of our knowledge, the papers about entropy stabilized intermetallic compounds are limited to those already cited and discussed in the submitted manuscript. Moreover, our manuscript is the first paper reporting additive manufacturing of the high/medium entropy intermetallic alloys. However, we added some more sentence about the novelty and current state of the topic in the last paragraph of the introduction.
- In my opinion Conclusionschapter should be a little changed. In this chapter there are no summary of all significant information obtained by the Authors and written in the chapters 2 and 3.
Thank you for your comment. We've added some clarifying points to the «Conclusion» section. But we did not greatly increase it, so as not to overwhelm the information from the "Discussion" section.
- In the list of references I found 5 papers of the Author of reviewed paper. Please indicate the differences in the studies presented in the cited article and their relation to the presented topic.
Answer: The reviewer is right. References [1] and [2] are two our previous review papers about the high entropy alloys, which cited here to show the recent interest, development, and range of application of high entropy alloys. Reference [1] is about the application of HEAs in nuclear reactors, and reference [2] discusses the recent development in the high entropy alloys for catalytic and functional applications. References [12] and [13] are our previous works on the additive manufacturing of HEAs. Reference [12] is a review paper indicate the current state of AM of high entropy alloys. Reference [13] show an example of HEA coating developed by laser cladding, a related topic to the present paper. We used reference [14] as an evidence to the effective diameter of electron in EDS analysis. Besides, we cited our recently published paper (reference [3]) to the manuscript, which discuss a novel application for HEA particles as reinforcement phase in the manufacturing process of matrix body drill bits.
Please prepare a literature review according to the guidelines of the Metals
Answer: Thank you for your comment. References of literature have been corrected in accordance with the requirements of the journal.
